# From Electronic Waste to 3D-Printed Product, How Multiple Recycling Affects High-Impact Polystyrene (HIPS) Filament Performances

**DOI:** 10.3390/ma16093412

**Published:** 2023-04-27

**Authors:** Edbert Wing Hanitio, Novan Rifky Lutfhyansyah, Balqis Mentari Efendi, Yati Mardiyati, Steven Steven

**Affiliations:** Materials Science and Engineering Research Group, Faculty of Mechanical and Aerospace Engineering, Institut Teknologi Bandung, Jl. Ganesha No. 10, Bandung 40132, Indonesia

**Keywords:** 3D printing, electronic waste, high-impact polystyrene, mechanical properties, recycling

## Abstract

The rapid growth of the electronics industry is producing excessive electronic waste. One of the common types of materials in electronic waste is high-impact polystyrene (HIPS). In this study, HIPS from electronic waste was recycled through an extrusion process and used as a 3D print filament. The effects of recycling on printability, physical properties, and mechanical properties in horizontal and vertical directions were examined. It was found that until the fourth-cycle, mechanical properties such as horizontal tensile strength, horizontal flexural strength, vertical flexural strength, and vertical impact strength were comparable with virgin commercial filament. In addition, the vertical flexural modulus in the fourth cycle increased by 77.28%. However, the density of recycled HIPs’ first to the fourth cycle slightly decreased by 10.6%, and the melt flow rate increased by 20.3%. It was also observed that until the third cycle, the effect of the reprocessing steps was insignificant on the defect of the 3D-printed product. In general, the experiments show various results, mainly in mechanical properties. Nevertheless, recycled HIPS filaments are comparable to or better than commercial ones in some cases. As a result, recycled HIPS filaments hold the potential to be considered as an alternative to other types of 3D print filaments.

## 1. Introduction

In the recent century, polymeric materials experienced a surge in popularity in the electrical and electronic equipment industry (EEE) due to their natural insulator properties. Consequently, the waste from the EEE industry, namely electronic waste (e-waste), grew as old appliances were replaced by newer ones [1]. According to a recent analysis by the United Nations, only 20% of the estimated 44.7 million metric tons (MMT) of e-waste was confirmed to be collected and effectively recycled [2]. Containing an estimate of 20–35% plastics, high-impact polystyrene (HIPS) and acrylonitrile butadiene styrene (ABS) were two of the most abundant plastics found in e-waste [3,4].

The abundance of these plastics can be taken advantage of by recycling. With recycling, the amount of plastic waste can be reduced while having the potential to create high-value products. The recycling process of plastic is not only an environmentally sustainable act but can also be viewed as an economically sustainable act [5,6,7]. One of the products that can be made by recycling plastics is a 3D-printing filament, which is essential in the additive manufacturing (AM) process and, more specifically, the fused deposition modeling (FDM) process. Additive manufacturing is a rapid prototyping technology that builds a structure layer by layer, following a computerized model [8]. The main advantages of the AM process are design flexibility and cost efficiency [9]. According to AM, FDM, better known as 3D printing, is frequently used to create prototypes or final products [10,11,12,13].

Several 3D printing filaments are widely used or commercially available, namely polylactic acid (PLA), ABS, HIPS, polyethylene terephthalate (PET), polypropylene (PP), high-density polyethylene (HDPE), and nylon. Most of those filaments have their specific usage save for PLA being the most generally used 3D-printing filament [5,14]. For instance, HIPS has good flow characteristics and impact resistance and is relatively cheap. On the other hand, HIPS filament is susceptible to wear and requires a high printing temperature. HIPS has characteristics similar to one of the widely used 3D-printing filaments, acrylonitrile butadiene styrene (ABS), yet is lighter in weight [7,15]. Xing et al. blended recycled ABS and recycled HIPS by the melt extrusion method using D-Glucose as a compatibilizing agent. With the addition of D-Glucose, the mixture of recycled ABS and recycled HIPS significantly increased in tensile strength, impact, and flexural strength [16]. While many studies have been conducted to improve various filament properties, there was little to no research paper investigating the influence of the recycling process in uncommon materials for 3D-printing filament, notably the recycled HIPS filament, despite HIPS having a non-biodegradable nature. Hence, the need for recycling HIPS is much higher than other polymers’ waste, such as PLA, which can degrade over time [17].

Nevertheless, materials transformation, such as the recycling process, induces material degradation. Moreover, through the mechanical recycling process of plastics, degradation has always been an issue due to the break of polymer chains [18,19]. Cruz et al. analyzed the degradation of PLA (polylactic acid) filament along five recycling cycles. It was demonstrated that there was a slight reduction in the mechanical properties through the cycles. The molecular weight of the polymer also decreased by about 47% after five recycling cycles, decreasing the tensile strength [20]. Polycarbonate also undergoes a decrease of 30% in tensile strength after being recycled ten times [21]. To our best knowledge, no study covered the recycling of HIPS into 3D-printing filament; however, some investigated the changes in HIPS’ properties after repeated recycling through injection molding. Vilaplana et al. reported that after nine reprocessing steps, virgin HIPS showed a total increase of 45% in the MFR value. Meanwhile, the value of stress at break and elongation at break showed an increase in the former and a progressive diminution in the latter [22]. Another finding from Bhilat et al. also reported the same occurrence of an increasing MFR value after six successive recycling cycles; however, there was disagreement regarding the tensile test result [23]. Therefore, this paper aims to investigate the effects of the reprocessing steps on the mechanical properties and printability of recycled filaments. Figure 1 shows the overview of the recycling HIPS process conducted in this study.

## 2. Materials and Methods

### 2.1. Raw Materials and Preparation

The raw materials (rHIPS-1) used in this study were obtained from Kitakaya Raharja Indonesia Inc., (Bekasi, Indonesia), in the form of chopped recycled HIPS (rHIPS) flakes in various sizes that are less than 2 cm, as shown in Figure 2. These flakes were washed with tap water and dish soap and dried in an oven at 60 °C for 1 h to obtain clean, recycled HIPS flakes. The clean, recycled HIPS flakes were considered first-cycle and would be extruded into second-cycle filament. After obtaining the second-cycle HIPS filament (rHIPS-2), the excess filament would be cut into pellets using a pelletizer at 50 rpm and cutting speed and at 15 rpm puller speed. The pellets were then used as the raw materials for extruding the third-cycle HIPS filament (rHIPS-3). The process was repeated and referred to as the reprocessing step until the fourth-cycle HIPS filament (rHIPS-4) was obtained. On the other hand, the commercially available 3D-printing filament chosen as the baseline in this study was from Shenzen Esun Industrial Co., Ltd., (Shenzhen, China). Table 1 summarizes the samples used throughout this study.

### 2.2. Filament Extrusion

The extrusion machine used in this study is a simple single-screw extrusion machine made by the Mechanical Production Laboratory and owned by the Materials and Metallurgy Laboratory, Institut Teknologi Bandung (ITB), Indonesia. The extrusion process parameters were set as follows: extrusion temperature at 195 ± 5 °C; screw speed at 19.7 rpm; puller speed at 15 ± 0.5 rpm. The cooling element used in the extrusion process was air from fans. One of the resulting recycled HIPS filaments can be seen in Figure 3.

### 2.3. 3D Printing Process

The diameter of recycled HIPS 3D-printing filament was 1.785 ± 0.145 mm. In this paper, FlashForge Adventurer 3 was used as the 3D printer for printing the 3Dbenchy, calibration cube, tensile specimens (ASTM D638 type IV), bending specimens (ASTM D790), and Izod impact specimens (ASTM D256). The prepared specimen models were sliced in FlashPrint 5 software to set the printing parameters. The printing parameters used in this study for the 3Dbenchy, horizontal mechanical specimens, and vertical mechanical specimens are summarized in Table 2. In addition, all mechanical specimen models are further specified into two types in Figure 4.

### 2.4. Density Test

The density of the rHIPS samples was determined according to Test Method B in ASTM D792. This particular test used a 25 mL pycnometer and 96% ethanol as the apparatus and tester fluid.

### 2.5. Melt Flow Rate Test

The melt flow rate test was conducted using the Presto Melt Flow Index Tester at Kitakaya Raharja Indonesia Inc. The testing temperature was set at 200 °C, and the testing load was set to 5 kg, following ASTM D1238.

### 2.6. Gel Permeation Chromatography (GPC)

A gel permeation chromatography test was performed using a Shimadzu LC-20 system equipped with a refractive index (RI) detector at the Serpong Advance Characterization Laboratory to determine the molecular weight of each recycled HIPS filament. The solvent used for the test is tetrahydrofuran (THF), with polystyrene (PS) as the standard material.

### 2.7. Fourier Transform Infrared Spectroscopy (FTIR)

Fourier transform infrared spectroscopy (FTIR) was conducted on each rHIPS specimen using the Bruker FTIR ALPHA II spectrometer (ATR mode) at the Materials and Metallurgy Laboratory, ITB, Indonesia. Each specimen was first cleaned using wipes and technical-grade alcohol before placing them on top of the ATR crystal. The scanning range chosen for the test was between 3500–500 cm^−1^ with 50 average scans and a spectral resolution of 2 cm^−1^.

### 2.8. Tensile Test

The tensile tests were conducted on all the tensile specimens, according to ASTM D638, using the Dynatech Tensilon RTF-1310 at the Materials and Metallurgy Laboratory, ITB, Indonesia. The test included two different types of specimens for measuring HIPS’ horizontal and vertical infill tensile strengths, which both were printed in the specimen-type IV shape. The 3D printing parameters for the mechanical specimens in both infill directions are stated in Table 2. The test speed was set at 5 mm/min, and the load cell rating was 10,000 N.

### 2.9. Bending Test

The bending tests were conducted on all bending specimens, following ASTM D790, using the Dynatech Tensilon RTF-1310 at the Materials and Metallurgy Laboratory, ITB, Indonesia. This test was conducted on the horizontal and vertical infill specimens, which followed the printing parameters in Table 2 to measure the overall HIPS bending strength. The bending test speed was set at 1.4 mm/min.

### 2.10. Izod Impact Test

The Izod impact tests were conducted on all the impact specimens owing to ASTM D256 using the Digital Presto IZC-222 Impact Tester at Kitakaya Raharja Indonesia Inc. The specimens were modeled after the ASTM D256 description. This test was also conducted with two types of specimens: one was the horizontal infill specimens, and the other was the vertical infill specimens. The horizontal and vertical 3D printing parameters are stated in Table 2.

### 2.11. Visual Inspection

Defect examinations were conducted on each 3D-printed sample to evaluate the printability of the 3D-printing filaments. The defect examinations were conducted by inspecting the HIPS 3Dbenchy specimens printed with the parameters in Table 2.

## 3. Results

### 3.1. The Effects of Reprocessing Steps on Density, Melt Flow Rate, and Molecular Weight

The density test was conducted to discover changes in the rHIPS molecular weight after being reprocessed multiple times. Figure 5a shows a notable density decline from rHIPS-1 to rHIPS-2 and a slight decline for the following reprocessing steps. Meanwhile, Figure 5b shows an increasing trend for the MFR values after each reprocessing step, especially from rHIPS-3 to rHIPS-4. The MFR test was intended in order to discover changes in rHIPS viscosity, which can be further linked to the PDI value in the GPC test. These trends of decreasing density and increasing MFR values were due to the degradation of the polymer chains that caused a chain scission phenomenon, leading to a reduction in the HIPS molecular weight and higher PDI values. Furthermore, those statements were backed up by the GPC results that can be found in Table 3.

Based on the MFR result in Figure 4b and PDI values in Table 3, it was found that the degradation phenomenon starts to heavily affect the HIPS filament viscosity in the fourth cycle, which is in line with the 35.18% increase in PDI values from the third-cycle to the fourth-cycle HIPS filament. Because of thermal reprocessing, the increasing PDI values represent the chain scission phenomenon in rHIPS chains. As a result, there were shorter chains than before, which increased the polymer’s flowability of the polymer, thus lowering the viscosity.

The GPC results in Table 3 show a decreasing molecular weight trend as the reprocessing steps increase. For example, the rHIPS-4 sample shows a decline of 21.88%, from 1691 g/mol to 1321 g/mol in the number average molecular weight (Mn) compared to rHIPS-2, thus validating the decreasing trend in the density test results. However, there is a slight increase in molecular weight between the rHIPS-2 and rHIPS-3 samples, which can be assumed that the thermal oxidative degradation process is incomplete.

By definition, HIPS is a copolymer that comprises polystyrene that is grafted on butadiene-based rubber surfaces, such as polybutadiene (PB) [24]. Therefore, when HIPS is undergoing thermal oxidative degradation, it starts in the rubbery phase of polybutadiene (PB) before spreading to the polystyrene (PS) because the C-H bond of PS is more stable than the C-H bond of PB [25]. However, it was found that PS also underwent thermal degradation simultaneously, which will be further explained in the next subchapter. This incomplete thermal oxidative degradation process, as seen in Figure 6, is initiated by hydrogen abstraction, which leads to the making of hydroperoxide radicals that may have a higher molecular weight than the starting compound; hence, the slight increase in molecular weight [26,27]. The resulting degradation mechanism can lead to two outcomes, as seen in Figure 7: chain scission and crosslinking.

In the rHIPS-4 samples, one could argue that the thermal oxidative degradation process had been completed, resulting in chain scission of the HIPS chain; therefore, the higher PDI value. An increase of 28.62% in the PDI value from rHIPS-2 to rHIPS-4 can also be linked to the MFR experimental findings that show an increase in the MFR value in rHIPS samples after experiencing more reprocessing steps, which meant the flowability of the chains had been improved due to having a more non-uniform chain length via the chain scission phenomenon. There are also several possibilities of crosslinking between the chains due to the produced free radicals [27]. These combinations of crosslinking possibilities are shown in Figure 7 and can affect the rHIPS molecular weight.

### 3.2. The Effects of Reprocessing Steps on FTIR Spectroscopy

The influence of the reprocessing steps on the rHIPS chains can be further discussed by referring to the FTIR spectroscopy results in Figure 8. In order to obtain a more comparable result of the structural changes in recycled HIPS caused by the reprocessing steps, the reference peak for absorbance normalization was set at 1492 cm^−1^, according to the findings of Vilaplana. The peak at 1600 cm^−1^ was caused by the aromatic C=C stretching vibration, whereas the peak at 1449 cm^−1^ was related to the aromatic ring stretching vibrations [22]. From the FTIR spectra, the peak corresponding to the polybutadiene microstructure cis-1,4 from the polybutadiene phase can be seen at around 1650 cm^−1^ [29]. As was mentioned in the previous subchapter, the degradation of HIPS starts not only within the polybutadiene phase but also within the polystyrene phase at the same time. It was proven by the quite significant increase in C-H bond intensity of the rHIPS samples at around 3000–2840 cm^−1^ after experiencing more reprocessing steps. That would not have been possible if only the polybutadiene was experiencing thermal oxidative degradation and, thus, hinted at the possibility of polystyrene degrading simultaneously. As a result of the degrading polybutadiene, the C=C bonds of polybutadiene experienced a decrease in intensity with each reprocessing step at around 966 cm^−1^.

### 3.3. The Effects of Reprocessing Steps on Horizontal and Vertical rHIPS Tensile Properties

Figure 9 summarizes the experimental results of the HIPS tensile strength and elastic modulus, while Figure 10 presents a preview of the 3D-printed tensile specimens in both infill directions. The rHIPS specimen with the highest horizontal tensile strength of 21.48 MPa was in the fourth cycle, and the lowest horizontal tensile strength of 16.02 MPa was in the second-cycle HIPS filament. Meanwhile, the rHIPS specimen had the highest vertical tensile strength of 15.69 MPa and the lowest vertical tensile strength of 4.18 MPa. There is a noticeable difference in tensile strength between the 3D-printed virgin HIPS, around 19.84 MPa, and the experimental results [30].

All horizontal infill specimens have higher tensile strengths than the vertical ones due to the difference in infill to tensile load directions. In the horizontal specimens, the tensile load direction is flat to the infill directions, making it more resilient to failure. On the other hand, the vertical specimens’ infill directions are perpendicular to the tensile load directions, making them rely heavily on the interlayer bonding strength.

Based on horizontal tensile strength in Figure 9a, it was found that the rHIPS tensile strength increased with each addition of the reprocessing step. It is caused by the decreasing total infill gap in the 3D-printed specimens as the rHIPS viscosity and melt strength decrease. This phenomenon has been schematized in Figure 11, highlighting the differences between the second- and fourth-cycle 3D-printed samples. Having a lesser infill gap equals more contact area between the infill and, thus, a higher tensile strength. Another explanation is the temporary crosslinking possibilities resulting from the degradation mechanism, as seen in Figure 7, which also helped increase the tensile strength. However, the same cannot be said for the vertical tensile strength in Figure 9b when the reprocessing steps increased. There is only a slight difference in value between the second, third, and fourth cycles when the standard deviation is taken into account. Therefore, their strength was not that affected by the infill gap. In the vertical infill direction, the specimen’s tensile strength relied heavily on the interlayer bonding strength, thus making their strengths similar.

The elastic modulus of the horizontal HIPS specimens can be found in Figure 9c, in which the highest horizontal elastic modulus of 2558.27 MPa was in the fourth cycle, and the lowest horizontal elastic modulus, 1514.17 MPa, was in the second-cycle HIPS filament. When compared to the elastic modulus of 3D-printed virgin HIPS, which holds for 1210 MPa, the recycled HIPS filaments show a much better performance for every reprocessing step [30]. Next, the elastic modulus of the vertical HIPS specimens can also be found in Figure 9d, where the highest vertical elastic modulus of 2557.13 MPa was in the second cycle, and the lowest vertical elastic modulus, 1298.7 MPa, was in the commercial HIPS filament. The elastic modulus experimental results have a similar trend to the tensile strength experimental result for the same reason explained in the previous paragraph.

Overall, the horizontal tensile test increased the tensile strength and elastic modulus as the reprocessing steps increased. Although the fourth-cycle HIPS filament boasted the highest value, further testing will be required to find out the limit of reprocessing on recycled HIPS filament tensile strength. Meanwhile, the vertical flexural test demonstrated that a lower tensile strength and elastic modulus are expected with the repetition of the reprocessing steps. Considering the lower viscosity of the fourth-cycle HIPS filament, the lower melt strength was also attributed to weaker interlayer bonding, thus lowering the tensile strength and elastic modulus altogether.

### 3.4. The Effects of Reprocessing Steps on Horizontal and Vertical rHIPS Flexural Properties

The experimental results of the HIPS horizontal flexural strength and flexural modulus are shown in Figure 12a,c. The highest horizontal flexural strength and modulus were found in the fourth cycle with means of 34.35 MPa and 2095.88 MPa. Meanwhile, the lowest horizontal flexural strength and flexural modulus were found in the third-cycle HIPS filament with means of 24.09 MPa and 1066.5 MPa. The initial decrease of flexural strength from the second to the third cycle was caused by degradation in the rHIPS chains, as discussed in the earlier subchapter regarding the MFR and GPC tests. However, as the degradation mechanism continues, a temporary crosslink between the rHIPS chains could form, resulting in the sudden increase of flexural strength and modulus from the third to the fourth cycle. The crosslinking limited the rHIPS chain’s movement; hence, stiffer chains with higher bending strength were established. Moreover, the flexural strength of rHIPS can be further improved by implementing the optimal 3D print parameters, as observed in a recent experiment by Tanoto, which utilized different factors, including the orientation position, fill pattern, fill density, and layer thickness [31].

From Figure 12b,d, the fourth-cycle HIPS filament held the highest vertical flexural strength and modulus of 22.79 MPa and 1842.85 MPa. On the contrary, the second-cycle HIPS filament held the lowest vertical flexural strength and modulus of 17.26 MPa and 1025.33 MPa. Although the same trend of decreasing flexural strength and modulus from the second to the third cycle and increasing flexural strength and modulus from the third to the fourth cycle matches with the horizontal flexural test result, the value changes were not as significant because the vertical bending specimens relied more on the interlayer bonding strength. Then, the increase in the flexural strength and modulus of the fourth-cycle HIPS specimens can also occur thanks to the lower viscosity, hence the larger contact area between the infills. In general, the horizontal flexural strength is higher than the vertical ones because more horizontal infills withstood the bending load than the vertical infills, as seen in Figure 13. Figure 14 shows the 3D-printed bending specimens in two different infill courses.

### 3.5. The Effects of Reprocessing Steps on Horizontal and Vertical rHIPS Impact Properties

The experimental results of the impact testing on the HIPS specimens are summarized in Figure 15, and the 3D-printed impact specimens can be seen in Figure 16. The second-cycle HIPS filament had the highest impact strength at 30.34 kJ/m^2^ (horizontal infill) and 6.37 kJ/m^2^ (vertical infill). Meanwhile, the fourth-cycle HIPS filament held the lowest impact strength with means of 8.33 kJ/m^2^ (horizontal infill) and 4.76 kJ/m^2^ (vertical infill). The experiment result of rHIPS-2 shows that with increasing the reprocessing steps, the overall horizontal impact strength of the HIPS filament decreased while the vertical impact strength only marginally decreased. Continuing from the MFR test result in Figure 5b, which had a trend of increasing value after each reprocessing step, the lower viscosity gave way to the rHIPS filament to fill the gap between the infill. With fewer infill gaps, the overall porosity in the impact specimens decreased; therefore, there was little to no slip motion that could help absorb the impact load.

Consequently, the rHIPS-4 horizontal impact specimens absorbed the least impact load among the others. This statement also applies to the vertical impact results, however, with a lesser magnitude, as their strength relies more on interlayer bonding. Nonetheless, the rHIPS-4 specimens were still comparable to the commercial HIPS impact strength. Furthermore, research conducted by Patterson provides the 3D-printed virgin HIPS impact strength data of 14.1 kJ/m^2^, which are lower than rHIPS-2 and rHIPS-3 but higher than rHIPS-4 [32]. This comparison consolidates the potential use of recycled HIPS filament in products that require high-impact absorbance. Figure 17 was added to help visualize the difference between the horizontal and vertical infill impact specimens when subjected to an impact load.

### 3.6. The Effects of Reprocessing Steps on the Visual of 3D-Printed HIPS Product

At the time of writing, the 3Dbenchy model was anonymously accepted as one of the standards in the 3D printing community, with increasing research authors mentioning it in scientific articles; for example, Rousseau and Novak [33,34]. The visual examinations were conducted by analyzing the visual defects on 3D-printed 3Dbenchy using commercial and rHIPS filaments. The results were then summarized in Table 4. All 3Dbenchy specimens had some defects, such as ringing, blobs, and zits, and overhang. Meanwhile, Figure 18 shows that the rHIPS-4 3Dbenchy had an additional stringing defect near the back when compared to the rest. This defect was caused by the rHIPS-4 filament having a higher MFR value, leading to some of the filament at the hot end of the 3D printer oozing out while the extruder was moving around. Overall, the difference in the visual defects of the 3D-printed rHIPS-2 and rHIPS-3 specimens is negligible compared to the 3D-printed commercial HIPS specimens. However, the rHIPS-4 filament shows a downgrade by having a stringing defect.

## 4. Conclusions

This study successfully produced low-cost recycled HIPS filaments from electronic waste, which is non-biodegradable and abundant. All the recycled filaments possessed good mechanical properties and printability except for the fourth cycle, which had slightly worse printability than the others due to an increased melt flow rate. Furthermore, the melt flow rate and density of recycled HIPS filaments were somewhat affected by the thermal-oxidative degradation due to reprocessing. The experimental results show that the rHIPS-4 filament has the highest horizontal tensile strength of 21.48 MPa, while rHIPS-3 has the highest vertical tensile strength of 15.69 MPa. For the highest flexural strength, rHIPS-4 boasted the highest in both infill directions, 34.35 MPa for horizontal and 22.79 MPa for vertical. Lastly, regarding the highest impact strength, the rHIPS-2 filament holds the record of 30.34 kJ/m^2^ in the horizontal and 6.37 kJ/m^2^ in the vertical infill directions. Considering recycled HIPS filaments’ mechanical and printability properties, the total reprocessing steps can still be pushed to a greater extent. In this case, this is until the recyclability limitation of rHIPS as a 3D-printed filament is reached. In conclusion, HIPS from electronic waste can be recycled into 3D-printed filament until the fourth cycle, with comparable or even better mechanical properties when compared to commercial ones.

## Figures and Tables

**Figure 1 materials-16-03412-f001:**
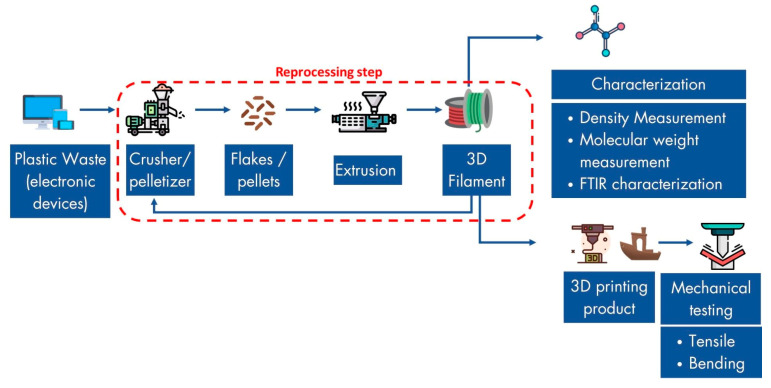
Summary of the main process of HIPS used in this study.

**Figure 2 materials-16-03412-f002:**
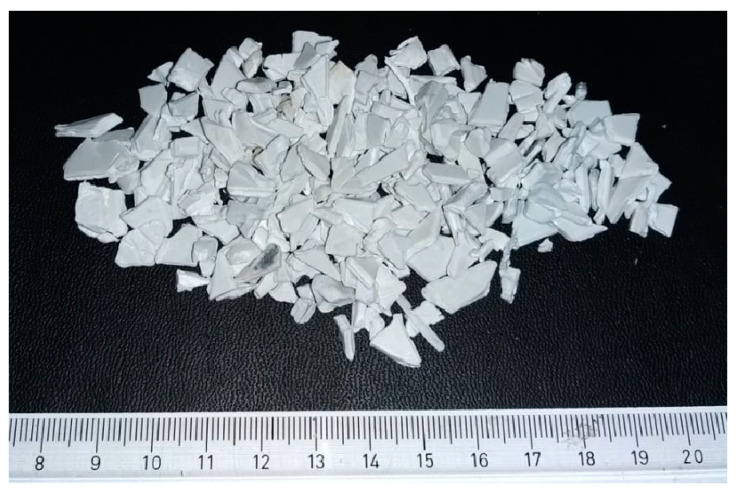
Recycled HIPS flakes in various sizes.

**Figure 3 materials-16-03412-f003:**
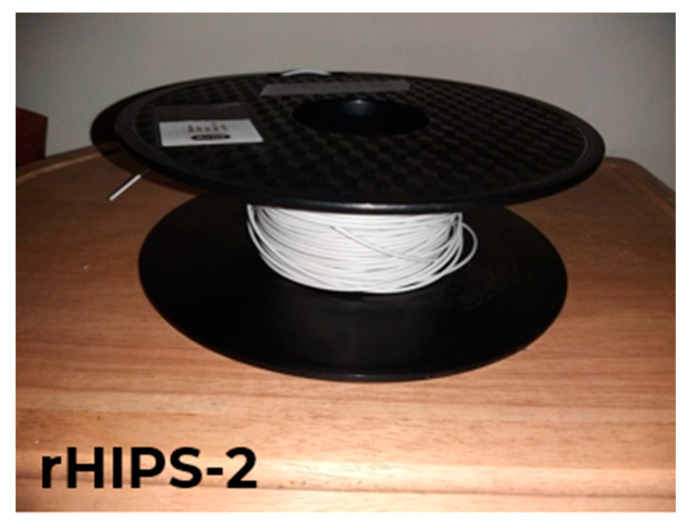
Recycled HIPS filament from the extrusion process.

**Figure 4 materials-16-03412-f004:**
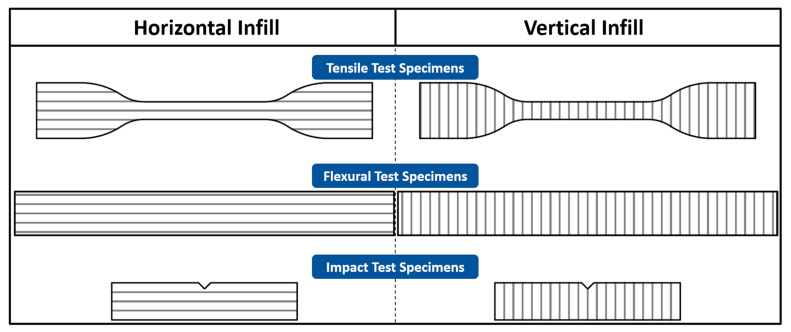
Tensile, flexural, and impact specimens with horizontal and vertical infill directions.

**Figure 5 materials-16-03412-f005:**
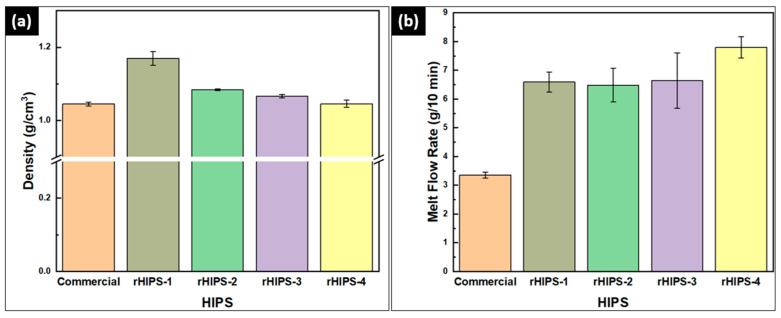
Density (**a**) and MFR (**b**) test results of commercial and rHIPS filaments.

**Figure 6 materials-16-03412-f006:**
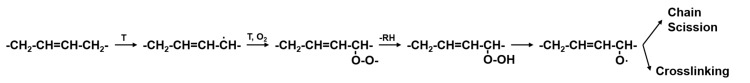
Thermal oxidative degradation mechanism of polybutadiene. Adapted with permission from Ref. [26]. 2012, Elsevier.

**Figure 7 materials-16-03412-f007:**
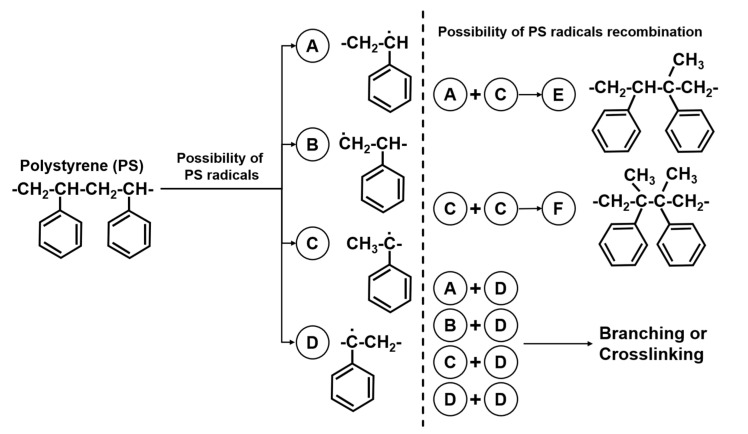
Crosslinking possibilities of PS chains. Adapted with permission from Ref. [28]. 2005, Elsevier.

**Figure 8 materials-16-03412-f008:**
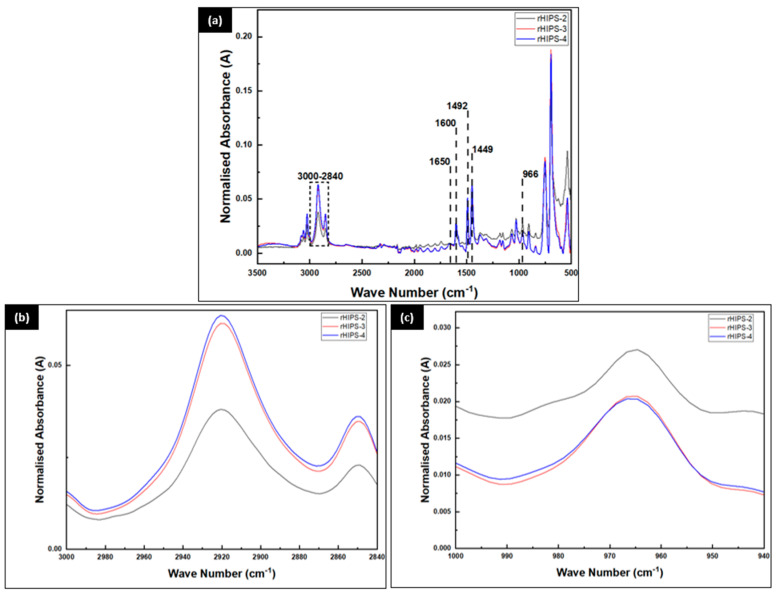
FTIR spectra of rHIPS samples: (**a**) 3500–500 cm^−1^; (**b**) 3000–2840 cm^−1^; (**c**) 1000–940 cm^−1^.

**Figure 9 materials-16-03412-f009:**
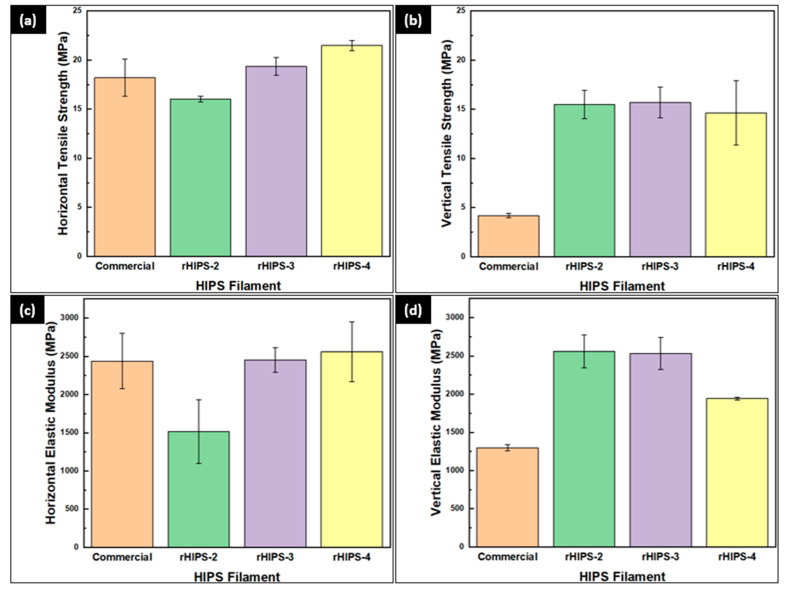
The tensile test result of 3D-printed HIPS specimens: (**a**) horizontal tensile strength; (**b**) vertical tensile strength; (**c**) horizontal elastic modulus; (**d**) vertical elastic modulus.

**Figure 10 materials-16-03412-f010:**
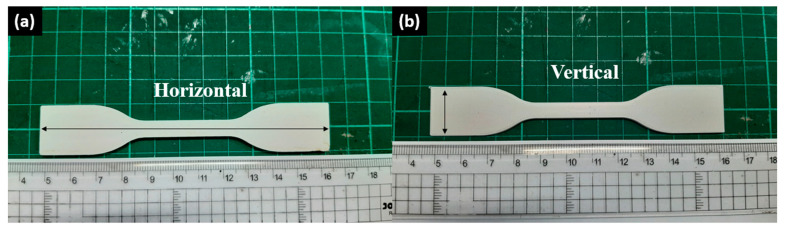
3D-printed tensile specimens: (**a**) horizontal infill; (**b**) vertical infill.

**Figure 11 materials-16-03412-f011:**
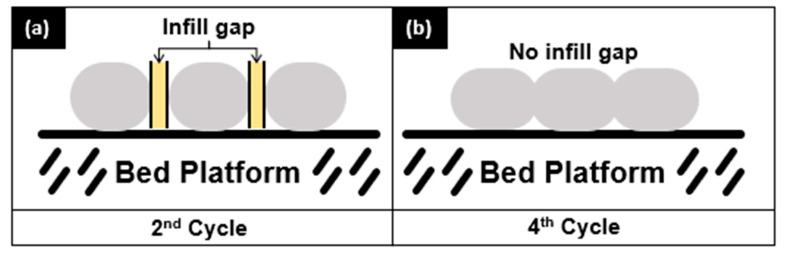
Scheme of infill gap in 3D-printed specimens: (**a**) 2nd cycle; (**b**) 4th cycle.

**Figure 12 materials-16-03412-f012:**
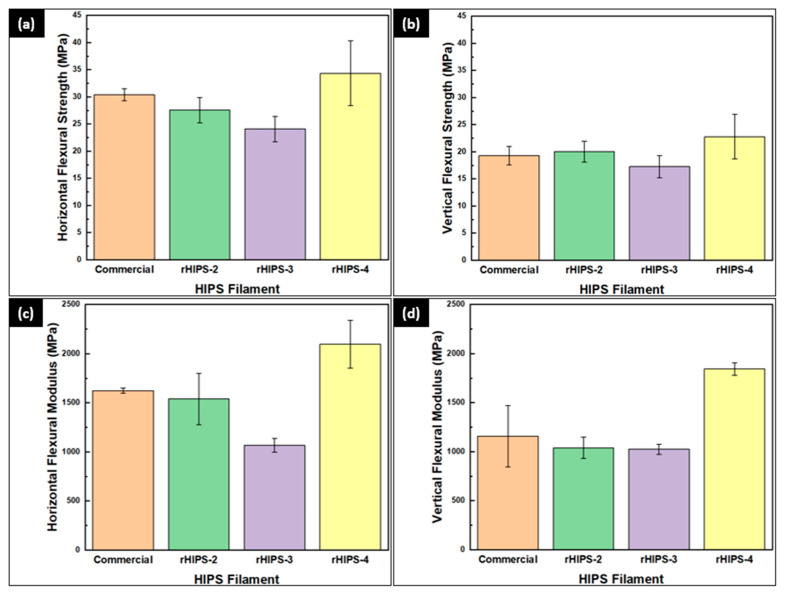
Flexural test results of 3D-printed HIPS specimens: (**a**) horizontal flexural strength; (**b**) vertical flexural strength; (**c**) horizontal flexural modulus; (**d**) vertical flexural modulus.

**Figure 13 materials-16-03412-f013:**
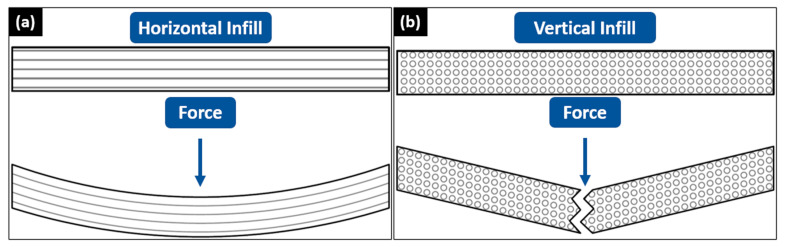
Front view of (**a**) horizontal and (**b**) vertical infill test specimens when subjected to a bending load.

**Figure 14 materials-16-03412-f014:**
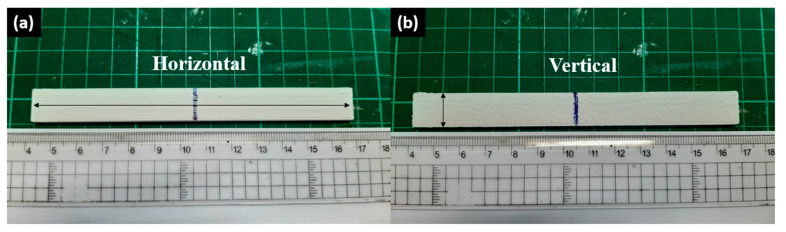
3D-printed bending specimens: (**a**) horizontal infill; (**b**) vertical infill.

**Figure 15 materials-16-03412-f015:**
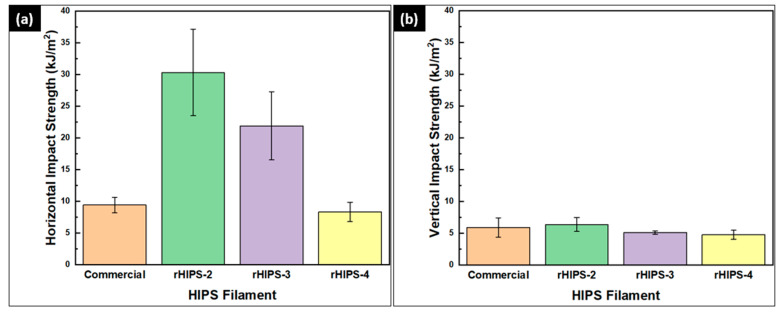
Impact test results of 3D-printed HIPS specimens: (**a**) horizontal impact strength; (**b**) vertical impact strength.

**Figure 16 materials-16-03412-f016:**
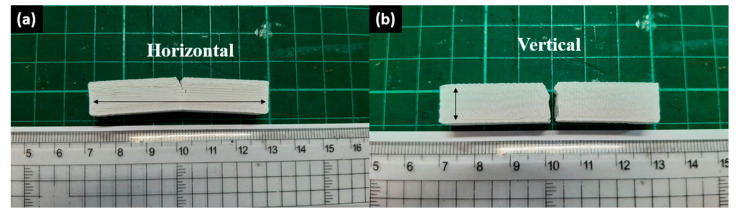
3D-printed impact specimens: (**a**) horizontal infill; (**b**) vertical infill.

**Figure 17 materials-16-03412-f017:**
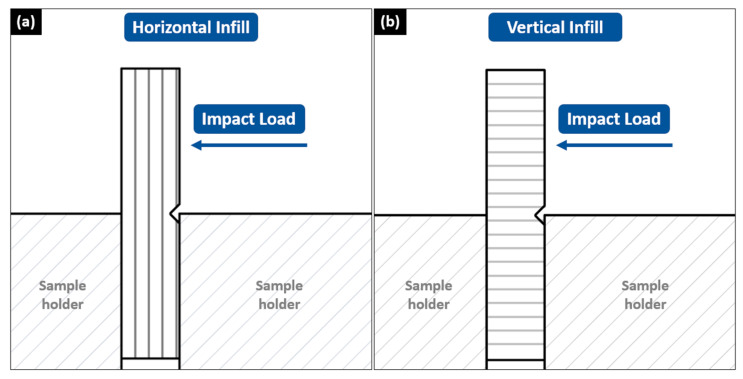
Front view of horizontal (**a**) and vertical (**b**) infill test specimens when subjected to an impact load.

**Figure 18 materials-16-03412-f018:**
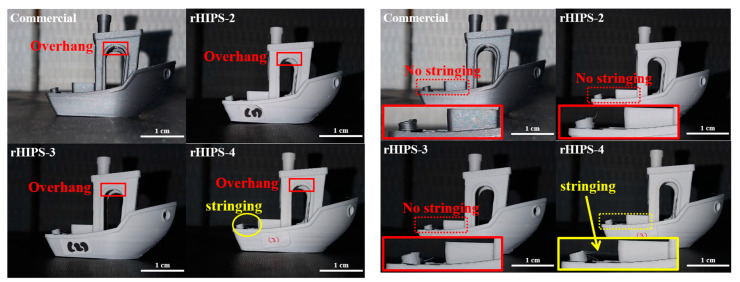
Side defect examination of all 3D-printed 3Dbenchy specimens using commercial and rHIPS filaments: (**left**) zoom out; (**right**) zoom in.

**Table 1 materials-16-03412-t001:** Sample codes for 3D-printing filament used in this study.

3D-Printing Filament	Sample Name
ESun HIPS	Commercial
Second-cycle HIPS	rHIPS-2
Third-cycle HIPS	rHIPS-3
Fourth-cycle HIPS	rHIPS-4

**Table 2 materials-16-03412-t002:** 3D printing parameters.

Parameters	3Dbenchy	Horizontal Infill	Vertical Infill
Nozzle temperature	230 °C	230 °C	230 °C
Bed temperature	100 °C	100 °C	100 °C
Print speed	50 mm/s	50 mm/s	50 mm/s
Layer height	0.18 mm	0.18 mm	0.18 mm
Infill density	15%	100%	100%
Infill pattern	Hexagon	Line	Line
Start angle	-	90°	0°
Cross angle	-	0°	0°

**Table 3 materials-16-03412-t003:** GPC test result of HIPS filament with the addition of 2,3 and 4 reprocessing steps.

Results	rHIPS-2	rHIPS-3	rHIPS-4
Mw (g/mol)	141,223	144,029	141,833
Mn (g/mol)	1691	1813	1321
PDI	83.5	79.45	107.4

**Table 4 materials-16-03412-t004:** Visual defects of 3D-printed 3Dbenchy specimens using commercial and rHIPS filaments.

Visual Defect	Commercial	rHIPS-2	rHIPS-3	rHIPS-4
Ringing	✓	✓	✓	✓
Blobs and Zits	✓	✓	✓	✓
Overhang	✓	✓	✓	✓
Stringing	✘	✘	✘	✘

## Data Availability

Data presented in this study are available on request from the corresponding author.

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
