# Peer review of "From Electronic Waste to 3D-Printed Product, How Multiple Recycling Affects High-Impact Polystyrene (HIPS) Filament Performances"

_materials, 2023, doi:10.3390/ma16093412_

Round 1
Reviewer 1 Report
The manuscript entitled „ From Electronic Waste to 3D-Printed Product, How Multiple Recycling Affect High-Impact Polystyrene (HIPS) Filament Performances” by Edbert Hanitio and coworkers presents the results of experimental studies of high-impact polystyrene to use in 3D printing. Due to the recent problems with the environment, the topic seems very interesting as the Authors try to use recycled plastic, obtained from the electronic industry.
Although the document presents interesting results and experimental methods that seem appropriate, the manuscript requires some changes.
- The experimental section requires some more information. Especially part dedicated to the FTIR spectroscopy lacks details of the experiment.
- What exactly is rHIPS-4? How do Authors define this material? It is not written.
- What is an explanation to present the same results in tables and in the graphs (see figure 10 and table 4, figure 12 and table 12 and figure 14 and table 6)? What new information we can obtain from the same data?
- What information should the reader receive from Figures 16, 17, and 18? It is almost impossible to observe any differences between the graphs.
- The summary part requires more information.
- Authors should carefully check the reference list and provide some small changes (reference 3, what “dan” means).
- The FTIR results should be once again carefully checked and it would be grateful to provide more information from the obtained results. For example, what is the origin of peaks around 1470 cm-1?
Taking into account the above-mentioned problems, I cannot recommend publishing the manuscript in its present form in the “Materials” Journal.
Author Response
Thank you very much for the comments and suggestions. Our response has been summarized in a Word file which can be found in the attachment.

Reviewer 2 Report
The aim of this work is to evaluate the properties of filament made of recycled HIPS. The goal of this study is very interesting in order to limit the materials quantity required for additive manufacturing.
This study is well conducted and all the conclusions are supported by experimental data. Characterizations are very completed.
However, I have some recommendations listed below:
Figure 3: There is no need to present all 3 pictures since no differences can be seen. If authors present only 1 picture, article will be clearer.
Line 107: How were the 3DBenchy printing parameters determined? Are they 3D printer supplier data?
Table 2: What is the layer height unit?
Figure 4: This figure is useless if figure 5 is presented.
Line 141: What is the speed rate? What is the load cell capacity?
Line 146: What is the speed rate?
Author Response
Thank you very much for the comments and suggestions. Our response has been summarized in Word file, which can be found in the attachment.

Reviewer 3 Report
The authors studied the HIPS from electronic waste recycled through an extrusion process and used as a 3D print filament. The manuscript had an interesting topic and was well-written, however it could only be accepted with the following minor revisions:
1. The abstract of the paper remains vague; the summary/conclusion of the study was not found in the text.
2. It is advised at the beginning of the Introduction Section the authors should define briefly about additive manufacturing (AM) technology and its advantages. Therefore, it is recommended the authors can add the following papers as references; Surface characteristics of machined polystyrene with 3D printed thermoplastic tool. Materials, 13(12), 2729; Mechanical, thermal and physical characteristics of oil palm (Elaeis Guineensis) fiber reinforced thermoplastic composites for FDM–Type 3D printer. Polymer Testing, 120 (2023), 107972.
3. According to Figure 1, the text or labeling was tidy and hardly read.
4. Refer to line 93, “the extrusion machine used in this study is a simple single-screw”, what is the brand/model/manufacturer of this extruder?
5. 3D-printed specimens for horizontal and vertical infill were not shown in the paper. Strongly advised, that the authors can include a figure of those samples.
6. Refer to Section 2.10 Izod Impact Test, what is the value of impact energy (J) used in performing the impact test?
7. Section 3.3, 3.4. 3.5 and 3.6 was very poor in citing the previous studies to compare with the result of the present study (must be improved).
8. The conclusion could be improved by adding future studies, limitations, and implications for researchers.
9. Too many repetitive sentences/ words throughout the paper, e.g; horizontal and vertical (13), according to (9), etc. Please read the article carefully (check the obvious repeated words) and proofread it from beginning to end.
Author Response

(The authors gave the same response as above.)

Round 2
Reviewer 3 Report
All corrections have been carried out by the author.